# Optimization of Extraction Conditions to Improve Phenolic Content and In Vitro Antioxidant Activity in Craft Brewers’ Spent Grain Using Response Surface Methodology (RSM)

**DOI:** 10.3390/foods9101398

**Published:** 2020-10-02

**Authors:** Ana Isabel Andres, Maria Jesus Petron, Ana Maria Lopez, Maria Luisa Timon

**Affiliations:** Food Technology Department, School of Agricultural Engineering, University of Extremadura, 06007 Badajoz, Spain; mjpetron@unex.es (M.J.P.); alopeznf@alumnos.unex.es (A.M.L.); mltimon@unex.es (M.L.T.)

**Keywords:** brewers’ spent grain (BSG), solid-liquid extraction, phenols, antioxidant activity, Response Surface Methodology (RSM)

## Abstract

Extraction temperature, extraction time and liquid (water) to solid ratio were optimized in order to extract antioxidant phenolic compounds from brewers’ spent grain (BSG). The extracts were analysed for their total phenol content (TPC) and antioxidant activity was measured using three different methods: 2,2-diphenyl-2-picrylhydrazyl (DPPH) free radical, 2,2′-azino-bis(3-ethylbenothiazoline-6-sulphonic acid) (ABTS), and reducing power (RP) assays. All the parameters except extraction time promoted different efficiencies for the extraction of antioxidant phenolic compounds. TPC extraction was higher at lower temperatures and lower liquid/solid ratios up to a certain point. In this sense, a decrease in TPC with increasing liquid/solid ratios took place until a 16:1 ratio and a plateau was reached beyond that ratio. The highest DPPH activity was reported for 30–35 °C and 60–90 min extraction and 60–90 min extraction with a 25 mL/g ratio. ABTS values increased as the liquid to solid ratio decreased, being positively correlated with TPC (*R* = 0.788; *p* < 0.01). The highest RP was achieved at 30–33 °C extraction temperature and 10–14 mg/mL *v*/*w* ratio and at 116–120 min extraction and 16–17 mg/mL ratio. Gallic acid accounted for the majority of the phenolic compounds found, followed by hydroxyphenylacetic acid, epicatechin, and protocatechuic acid. Sinapic, 4-hydroxy benzoic, and syringic acids were also found in lower quantities. Coumaric, vanillic, ferulic, and caffeic acids were present in very small amounts. All the extracts contained phenolics and showed in vitro antioxidant activity, but the extracts obtained by using 30 °C, 121.9 min, and 10 mL/g liquid/solid ratio exhibited the highest content in TPC and antioxidant potential. The aqueous extraction of a potentially bioactive extract from BSG was demonstrated to be an efficient and simple method to recover these value-added compounds.

## 1. Introduction

Over the last several years, the global beer industry has experienced an outstanding increase in production and consumption, reaching up to 1.95 billion hectoliters and 357 million hectoliters, respectively [1]. In particular, the craft brewing industry has boomed during the last decade and the number of breweries has doubled in EU from 2013, currently estimated at 9,500 breweries [1].

Concomitantly, the production of by-products has also become a major economic and environmental problem for the beer industry. The brewing process generates several by-products, among which brewers’ spent grain (BSG) is the most abundant. BSG is generated during the mashing process, after separation from the wort. Producing one hectoliter of beer gives rise to 20 kg of BSG [2]. Assuming this ratio of 1:5, the calculated amount for 1.95 billion hL beer is 390 million tons of wet BSG annually [2].

BSG is high in protein and fibre (cellulose, hemicellulose, and lignin) and thus, it can be used as a feeding ingredient for composting or for biogas and energy production. It has also been used in small-scale applications in high-fibre breads and cookies [3]. In addition, BSG also contains a variety of mineral elements and other valuable compounds, such as polyphenols, with many potential industrial applications in foods and supplements, since they have shown to delay oxidative reactions and microbial growth [3].

Phenolic compounds are mostly accumulated in the husk of cereals, which is the main component of BSG, as well as part of the pericarp and seed coat layers that are obtained as residual solid material after the wort elaboration step [3]. Phenolic compounds can be found in both free soluble and insoluble forms, bound to cell wall materials [3]. Most free phenolics are flavanols, whereas bound phenolics are mainly phenolic acids [4,5]. Considering the increasing amount of BSG produced annually in the brewing industry, efforts should be made to valorize this agro-industrial by-product derived from breweries. Therefore, the extraction of bioactive-rich extracts for reusing them in the food industry or in other related sectors could be a promising solution [3]. In order to obtain potentially functional extracts from BSG, several attempts have been carried out with different techniques, such as supercritical fluid extraction (SCFE), microwave assisted extraction (MWAE), pressurized liquid extraction (PLE), hydrolisis, and liquid-liquid and liquid-solid extraction with solvents (polar and non-polar) [6]. Techniques and solvents are of utmost importance to maximize recovery selectivity. Solvent extraction is the most traditional technique, with methanol, ethanol and acetone being the most used solvents for BSG [6,7,8,9]. The addition of water to these organic solvents is a common practice, since it creates a more polar medium and may facilitate the extraction of compounds that are soluble in organic solvents and/or water, as phenolics [8]. However, those mentioned solvents show some disadvantages, like their toxicity, low biodegradability, high cost, the need for further separation stages, and their environmental hazards [10].

The use of the so called “green extraction techniques” and “green solvents” has gained much attention nowadays. In this sense, water is the “greenest solvent” conceivable. Water is easily available, cheap, promptly recycled, non-toxic, and non-flammable and its use as a solvent can also meet requirements from an environmentally friendly and economically viable perspective [10]. In addition, water is a polar solvent and it is suitable for extraction of polar compounds as phenols [6]. Water has been used as an alternative for the recovery of phenolic compounds from potato, apple pomace [11], and from fermented BSG [12].

However, the selection of the extraction parameters, such as temperature, stirring rate, extraction time, sample particle size, pH, and liquid/solid ratio can determinantly affect the recovery of phenols from crop by-product substrates [6].

The selection of the optimal extraction parameters can be approached in a “one variable at a time” way, but this methodology is extremely time consuming and potential interactions among variables and parameters are not considered at all [13]. Response Surface Methodology (RSM) is an effective statistical technique for optimizing complex processes. CCD (Central Composite Design), one type of RSM, is more efficient, it is easier to arrange and interpret the optimization experiments compared with other designs, and it has been widely used to optimize a great number of parameters [13,14].

To the best of our knowledge, there is a scarce number of studies focused on the optimization of the extraction of phenol compounds from BSG and even less using a “green solvent” as water. Therefore, a systematic approach was used to optimize extraction factors such as temperature, time, and liquid/solid ratio, leading to the maximum extraction of phenols and the highest in vitro antioxidant activity.

## 2. Materials and Methods

### 2.1. Raw Material and Chemicals

Brewer’s spent grain was kindly donated by a local brewery (Ballut, Badajoz, Spain). The malt blend for this beer was made up of 50% Pilsen type barley malt and 50% wheat malt. Proximate composition was 75 ± 0.20 g/100 g, 4.85 ± 0.19 g/100 g, 20.42 ± 0.12 g/100 g, and 3.2 ± 0.98 g/100 g of moisture, fat, protein, and ash percentages, respectively. The spent grains used in this research were obtained as a by-product from the brewing process of “Belona” craft beer. Samples were inmediately dried at 100 °C up until 94% dry matter, packed in polyethylene bags, and safely stored at room temperature until further analysis.

Folin-Ciocalteau reagent was procured by Fisher (Loughborough, UK). Sodium carbonate, gallic acid, potassium ferrocianide, and acetic acid were provided by Scharlau (Sentmenat, Spain). Ascorbic acid, 2,2-diphenyl-1-picrylhydrazyl radical was from SIGMA (Steinheim, Germany), 2,2′-azino-bis(3-ethylbenothiazoline-6-sulphonic acid (ABTS) was from Alfa Aesar (Kandel, Germany), and Iron (II) Chloride 4-hydrate was purchased from Panreac Química SAU (Barcelona, Spain). Phenol standards were provided by SIGMA (Steinheim, Germany). Other chemicals and reagents used were analytical grade and commercially available.

### 2.2. Selection of Variables and the Extraction Process

The extraction process was carried out at different temperatures between 30 and 50 °C (*X*_1_), times (*X*_2_), between 60 and 180 min and ratios liquid/solid (water/BSG, *v*/*w*) (*X*_3_), between 10:1 and 25:1 (see Table 1). The selection of these values was done on the basis of preliminary studies and previous studies on beer by-products [15]. The following dependent variables (responses) were measured: total phenolic content (TPC), DPPH radical scavenging ability assay (DPPH), Reducing Power (RP), and ABTS radical cation inhibition antioxidant assay (ABTS).

The dried BSGs were mixed with different volumes of water, at different temperatures and times of extractions according to Table 1. These values were based on the experimental design produced by Design Expert version 10.0 (Stat Ease, Inc., Minneapolis, MN, USA). The pH of all mixtures was set at 3 in order to stabilize phenolic compounds. Extraction was carried out in a water bath and under continuous stirring. After extraction, the mixtures were fast cooled and filtered using a sieve with a 45 µm pore size (Filtra, Badalona, España). The clarified liquid was then centrifuged at 3000 rpm for 10 min. Supernatants were stored at −80 °C until further analysis. Each extraction assay (run) was performed in triplicate.

### 2.3. Experimental Design

Optimization of the experiment was carried out using Response Surface Methodology (RSM) for the extraction of total phenol content (TPC) and in vitro antioxidant activity (DPPH, RP, and ABTS). Central Composite Design (CCD) consisted of 20 experimental runs including 6 at central points, 6 at axial, and 8 at factorial points (see Table 1).

The extraction variables included extraction temperature (*X*_1_, °C), extraction time (*X*_2_, min), and liquid to solid ratio (*X*_3_, mL/g, *v*/*w*) (see Table 1). The experimental data were fitted to a second-order polynomial model to obtain regression coefficients (β_0_). The generalized second-order polynomial model used in the response surface analysis is as follows:Y = β_0_ + Σ^k^_i = 1_ β_i_X_i_ + Σ^k^_i = 1_ β_ii_X_i_^2^ + Σ^k^_i < j = 1_ β_ij_X_i_X_j_,(1)
where Y is the response variable, Xi and Xj represent independent variables, *n* is the number of tested variables, and ε denotes error. β_0_, β_i_, β_ii_, and β_ij_ refer to the coefficients of constant, linear, quadratic, and interaction effects, respectively. 

To evaluate the predicted model on the response variable, an analysis of variance (ANOVA) with a 95% confidence level was carried out to asses the effect of each factor (temperature, time, ratio). In addition, the regression coefficient (*R*^2^), the *p*-value of the regression model, and the *p*-value of the lack of fit (LOF) were used to determine the fitness of the regression model. Optimal conditions were chosen considering the response surfaces (3D plots).

The optimized conditions were validated for the maximum total phenol content (TPC) and antioxidant activities (DPPH, RP, and ABTS), based on the values obtained using RSM. All the responses were determined under optimized conditions of the extraction. The experimental values were compared with predicted values based on CV % in order to determine the validity of the model. The profile of phenolic compounds was also analysed at optimized conditions by HPLC. The verification of the validity and adequacy of the predictive extraction (*n* = 5) was done, comparing predictions with those observed values using a two-sided t-test (*p* = 0.05).

### 2.4. Quantification of Total Phenolics 

The amount of total phenolic content in extracts was determined in triplicate according to the Folin-Ciocalteu procedure [16]. Results are expressed as mg gallic acid equivalents (GAE) per g of BSG.

### 2.5. Antioxidant In Vitro Assays

#### 2.5.1. Determination of DPPH Radical Scavenging Activity

The antioxidant activity of the extracts based on the scavenging activity of 2,2-diphenyl-2-picrylhydrazyl (DPPH) free radical was determined in triplicate by the method described by Broncano et al. [17]. The ability to scavenge the DPPH radical was expressed as the inhibition percentage and was calculated using the following formula:RSA (%) = [(A_control_ − A_sample_)/A_control_] × 100,(2)
where the A_control_ is the absorbance of the control and the A_sample_ is the absorbance of the extract. The same concentration of ascorbic acid (vitamin C) was used as a positive control.

#### 2.5.2. Reducing Power

The RP was determined in triplicate according to the method described by Broncano et al. [17]. Absorbance was measured at 700 nm. Ascorbic acid (vitamin C) was used as a positive control. The calibration curve of vitamin C that was used for the reducing power determination ranged from 0 to 1 g/mL (0, 0.1, 0.2, 0.4, 0.6, 0.8, 1). Results were expressed as mg ascorbic acid/g BSG.

#### 2.5.3. ABTS Radical Cation Inhibition Antioxidant Assay

The scavenging ability against 2,2′-azino-bis(3-ethylbenothiazoline-6-sulphonic acid) was determined in triplicate according to the method reported by Świeca et al. [18], with some modifications. ABTS free radical (ABTS^+^) was generated by the oxidation of ABTS with potassium persulfate. To obtain the ABTS stock solution, 7 mM solution of ABTS reacted with 2.45 mM potassium persulphate (final concentration) and the mixture was allowed to stand in the darkness for 16 h at room temperature. The ABTS^+^ working solution was prepared by diluting the ABTS^+^ stock solution with distilled water to obtain an absorbance of 0.7 ± 0.05 at 734 nm. Then, 0.4 mL of sample extract was added to 3 mL of ABTS^+^ working solution. The absorbance was measured at 734 nm after 30 min incubation. Vitamin C was used as the standard. The radical scavenging assay for all samples was expressed as a percentage of scavenging ABTS radical using the equation:ABTS scavenging effect (%) = (A_0_ − A_1_/A_0_) × 100,(3)
where A_0_ is the absorbance of the control sample and A_1_ is the absorbance of samples. The same concentration of ascorbic acid (vitamin C) was used as a positive control.

#### 2.5.4. Validation of the Model

The optimized conditions of extraction (time, temperature, and ratio liquid/solid) were validated for the maximum phenolic content and in vitro antioxidant activities (DPPH, RP, and ABTS) on the basis of the obtained values using RSM. All the responses were again determined under the optimized conditions of extraction. The experimental values were compared with those predicted by the model in order to asses its validity. The HPLC profiles of phenolic compounds were also determined at optimized conditions.

#### 2.5.5. HPLC Analysis

Phenolic compounds in the extracts obtained at optimum conditions were analyzed in triplicate using a HP1200 Liquid Chromatograph with a diode array detector (Agilent Technology, Palo Alto, CA, USA) and an Inertsil ODS-3 column (5.0 μm particle size, 4.6 mm × 250 mm) (GL Sciences Inc., Tokio, Japan) preceded by an Inertsil ODS-3 Guard Column (5.0 μm, 4.0 mm × 10 mm). Elution gradient was: 0 min, 5% B; 13 min, 11% B; 16 min, 13% B; 20 min, 14% B; 22 min, 15% B; 25 min, 20% B; 28 min, 25% B; 30 min, 30% B; 40 min, 5% B. Runtime was 40 min. Solvent A was 2.5% formic acid in water and solvent B was 2.5% formic acid in acetonitrile. Solvent flow rate was 1 mL/min and the injection volume was 25 μL. Phenolic compounds were identified by comparison of their retention times with those of standards (see Appendix A). The concentrations of individual phenolic compounds in samples were calculated using calibration curves (see Appendix A). Results were expressed as micrograms/gram of BSG (µg/g).

## 3. Results and Discussion

### 3.1. Fitting the Model

Optimization of the extraction process was carried out by applying second order polynomial equations. The model shows high significance and good fit with the experimental data (*R*^2^ values 0.89, 0.88, 0.90, and 0.96 for TPC, DPPH, RP, and ABTS, respectively) (see Table 2).

The regression coefficients for dependent variables were obtained by multiple linear regressions as shown in Table 2. A negative linear effect of temperature (*X*_1_) was significant for TPC, as well as time (*X*_2_) for DPPH and ratio *v*/*w* for both TPC and ABTS variables. The interaction effect of *X*_12_ (temperature and time) was found to be significant only for DPPH, the interaction effect of *X*_13_ was found to be significant only for RP, and finally, the interaction effect of *X*_23_ was found to be significant for both DPPH and RP variables. The quadratic effect of *X*_2_ (time) produced significant negative effect on DPPH. The quadratic effect of *X*_3_ (ratio *v*/*w*) was found to produce a significant effect on all variables except for DPPH. 

The ANOVA results for each response variable indicated that the three parameters on the model can explain the experimental variation for response variables as shown by the significant F-value for the model (see Table 2). The lack of fit test was found to be non significant, thus indicating that the model could adequately fit the experimental data for all the response variables (see Table 2).

### 3.2. Effect of the Extraction Variables on Total Phenolic Content (TPC)

The model showed high significant (*p <* 0.01) value with the experimental data. Analysis of variance (ANOVA) showed a negative linear (*p* = 0.001) (*X*_1_ and *X*_3_) and quadratic effect (*X*_3_^2^) (*p <* 0.01) on TPC content (see Table 2). There was not a significant effect of interaction among *X*_1_, *X*_2_, and *X*_3_ on the experimental data. The non-significant value of lack of fit (*F* = 1.24; *p >* 0.05) showed that the model is fitted to the spatial influence of the variables to this response with good prediction (*R*^2^ = 0.89) and that it is not significant compared to the pure error.

TPC values were in the same range as those reported by Meneses et al. [8], using water as a solvent as well. Regression analysis results (see Table 2) indicated that TPC extraction was higher at lower temperatures and lower liquid/solid ratios to a certain extent. Temperature and solid-liquid ratio were also found to play the most critical roles in extraction efficiency of phenolic compounds from grape by-products, [19], as well as in black rice [20]. The relationship between TPC and process variables is depicted in Figure 1a,b. The highest extraction of TPC took place within the range of 30–35 °C and 10–13 mL/g. Extraction time did not play any significant role in TPC extraction, which is in agreement with previous findings on wheat [21]. Therefore, it can be assumed that the shortest studied time (60 min) is long enough for the extraction of TPC in the current experiment.

As has already been mentioned, temperature is known to play an important role in the efficiency of phenols extraction [6]. The higher the temperature, the higher the compounds solubility and diffusion coefficients are and the lower the solvent viscosity and surface tension, as well as the weaker the phenolic-protein and phenolic–polysaccharide linkages are. This is even more important in grain, where a substantial amount of phenolics is bound to cell wall materials [22]. All these phenomena would ease the migration of phenolic compounds into the extraction solvent [6,14]. Conversely, phenolic compounds have been found to be denatured by chemical or enzymatic reactions beyond a certain temperature [19]. Some authors have reported that high temperatures could promote membrane denaturation and consequently, mobility of both solvent and solutes could be compromised as well as the extraction process [23]. Undesired compounds could also be dissolved under these conditions [6]. The detrimental effect of temperature on TPC extraction could have prevailed in the tested conditions of the present experiment. In this sense, the positive effect of temperature on phenol extraction was not so conclusive in cereal, as reported by Balli et al. [24] and Moreira et al. [25], who observed a negative effect of increasing temperatures on TPC extraction in BSG. Cacace et al. [7] observed that an increase in temperature from 40 to 74 °C resulted in lower phenol extraction yield from black currants. 

Regarding the ratio of liquid/solid (*v*/*w*), the higher this ratio, the lower the extracted TPC was, as can be inferred from the significant linear negative coefficient in Table 2. However, there was also a significant quadratic effect of liquid/solid ratio on TPC (*β* = 0.722; *p* < 0.01), indicating that the ratio showed both positive and negative effects on TPC, in agreement with Liyana-Pathirana et al. [21]. In this sense, the decrease in TPC as the ratio of liquid/solid increased took place up until a 16:1 ratio; beyond that ratio, a plateau was reached. Pompeu et al. [26] reported that a plateau in the mass transfer was reached at a solid-to-liquid ratio of 1:2 for TPC. Above that ratio, higher TPC was not extracted from fruit. Similarly, Gubta et al. [27] determined a maximum TPC in brewing waste when the liquid/solid ratio was the lowest. When optimizing the extraction of phenolics in black rice bran, Pedro et al. [20] concluded that the lowest liquid/solid ratio was the most suitable. Todaro et al. [28] observed that the increase in the liquid/solid ratio only improved the yield for phenols (anthocyanins) in the case of acidified methanol, but not for tartaric and malic acids aqueous solutions. In contrast, some authors observed that TPC increased as the solid/liquid ratio increased from 1:10 to 1:50 (g/mL) in fruit extracts, though organic solvents were used instead of water [7,8,9,10,11,12,13]. Ryu et al. [29] did not observe a significant effect of the solid/liquid ratio when applying water extraction in black soy beans. These different results can be ascribed to the different matrixes, extraction methods, and phenolic compounds in plant tissues, which make comparison quite difficult.

### 3.3. Effect of the Extraction Variables on the Antioxidant Potential of BSG Extracts

The antioxidant potential of the BSG extracts was determined by three methods based on different approaches, namely the 2,2-diphenyl-1-picrylhydrazyl (DPPH) and 2,2′-azino-bis(3-ethylbenothiazoline-6-sulphonic acid) (ABTS) radical scavenging method and the ferric reducing antioxidant power (RP) assay, which have been widely used in plant and by-product extracts.

Analysis of variance (ANOVA) showed significant negative linear and quadratic effects of time (*X*_2_ and *X*_2_^2^) (*p <* 0.01 and *p <* 0.001, respectively) on DPPH values (see Table 2). The interactions time-temperature (*X*_12_) and time-ratio (*X*_13_) were also significant in the model (*p <* 0.01). This is shown graphically in Figure 1c,d. It can be observed that at a certain time, temperature, and ratio, DPPH reached its highest value and then began to decline. The highest activity was reported from the range of 30–35 °C and 60–90 min (Figure 1c) and the range of 60–90 min and 25 mL/g (Figure 1d). 

In the case of ABTS results, only the linear and quadratic effects of ratio liquid/solid (*X*_3_) were significant for the model (*p <* 0.001) (Table 2). ABTS values increased as the ratio decreased, as shown in Figure 2a,b. This behaviour is similar to that explained for TPC, with these parameters being positively correlated (*R* = 0.788; *p <* 0.01) (Table 3). 

This positive correlation has also been reported by previous authors in wheat grains [9,10,11,12,13,14,15,16,17,18,19,20,21]. However, ABTS was not significantly correlated with DPPH (see Table 3), which is agreement with results reported by Zhang et al. [30], but is not consistent with results by Socaci et al. [9]. It is well established that DPPH and ABTS assays show several differences in their responses to antioxidants. In this sense, ABTS can be solubilized in aqueous and in organic solvents, in which the antioxidant activity can be measured due to the hydrophilic and lipophilic nature of the compounds in samples [31]. In contrast, DPPH can only be dissolved in organic solvents (especially in alcoholic solvents), not in aqueous ones, which is an important limitation when interpreting the role of hydrophilic antioxidants [31].

Table 3 shows Pearson’s correlation coefficient calculated for the responses variables. From this table, it can be inferred that DPPH values are not significantly correlated with TPC. Meneses et al. [8] also reported a very low and weak correlation between these two parameters in BSG. It should be pointed out that the Folin-Ciocalteau method can measure compounds other than phenols and final absorbance could also be the result of other reduction reactions, such as those with ascorbates and thiols [32]. On the other hand, the antioxidant activity of a plant extract cannot only be related to its total phenolic content. Other specific phenolic compounds such as flavonoids, tannins, proanthocyanidins, and amino phenolic compounds can play a more important role in antioxidant activity, as observed by Meneses et al. [8] in BSG.

As for reducing power (RP) is concerned, only the interactions ratio-temperature (*X*_13_), ratio-time (*X*_23_), and the quadratic effect of ratio were significant in the model (*p <* 0.01). The highest RP was achieved from the range of 30–33 °C and 10–14 mg/mL and at 116–120 min and 16–17 mg/mL. This is shown graphically in Figure 2c,d. Reducing power and radical scavenging activity (DPPH) were positively correlated (*p <* 0.05) (see Table 3), which is in accordance with results by Meneses et al. [8], but those were not correlated with TPC (*p >* 0.05). 

As a conclusion, on the basis of these results, the model is statistically significant and offers solid answers as a function of independent variables. Therefore, it can be a useful tool for statistical process optimization.

### 3.4. Optimization of the Extraction Parameters and Model Validation

The optimal conditions were determined by maximizing the desirability of the responses using Design Expert Software Version 10.0 (Stat-Ease, Inc., Minneapolis, MN, USA). Ideally, the maximal desirability should be at the maximum concentration of TPC and the highest values for DPPH, RP, and ABTS. These optimal conditions were used for the extraction process and later, the responses were determined and validated according to the abovementioned procedure. The optimal conditions were 30 °C, 121.9 min, and 10 mL/g liquid/solid ratio. The obtained desirability was 0.828. Under these optimal conditions, the experimental values were in agreement with the predicted values, with a coefficient of variation (CV) ranging from 0.6 to 11.6 (see Table 4). This parameter describes the extent to which the data were dispersed.

### 3.5. HPLC Analysis of Phenolic Compounds 

Qualitative and quantitative analysis of phenolic compounds from the extract obtained at the optimized conditions (30 °C, 121.9 min, and 10 mL/g liquid/solid ratio) were performed using high performance liquid chromatography (HPLC) (see Appendix A). Results are shown in Table 5. Total amount of identified individual phenols reached 178 μg/g BSG, which is substantially lower than the result obtained by Folin-Ciocalteau (5.42 mg gallic acid/g BSG). The F-C assay does not only measure phenols, but could also react with other antioxidants besides phenols [33], such as proteins, carbohydrates, amino acids, nucleotides, thiols, unsaturated fatty acids, vitamins, amines, aldehydes, and ketones [32]. In this sense, some authors have suggested that measurement of phenols by the F-C method could possibly give too high an estimate of phenolic content [34].

Gallic acid accounted for the majority of phenolic compounds in this research, followed by hydroxyphenylacetic acid, epicatechin, and protocatechuic acids. Sinapic, 4-hydroxy benzoic, and syringic acids were also found in lower quantities. Coumaric, vanillic, ferulic, and caffeic acids were present in very small amounts. This phenolic profile differs substantially from results reported by Masisi et al. [35] and Stefanello et al. [36] in wheat grains, wheat bran, barley [37], and in general in BSG, where hydroxycinnamic acids, ferulic acid, coumaric acid, sinapic acid, and caffeic acid were the main phenolic compounds identified [5,38,39]. Other authors identified 4-hydroxy benzoic and protocatechuic acids as the most quatitatively important in BSG extracts using various solvents mixtures. 

The wide range in cereal varieties, malting and mashing conditions, and type and quality of secondary raw materials added in the brewing process [3] surely account for the reported differences to a certain level. Nevertheless, undeniably, differences among quantified phenols can be ascribed to the fact that the aforementioned authors used organic solvents alone or mixed with water, which would increase the extraction of phenol compounds [8]. According to some authors, the addition of water to organic solvents creates a more polar medium and may facilitate the extraction of chemicals that are soluble in organic solvents and/or water [8]. It should be underlined that a substantial amount of phenolic compounds associated with cereal grains, ranging from 54.6% to 75% [40], are in insoluble bound forms. However, there are several food processes that enhance the liberation of bound phenolics and that is important to consider in the case of BSG, such as fermentation and malting, as well as thermomechanical processes, which takes place in the current extraction process [40].

Despite this different profile, it can be assumed that the antioxidant activity of the extract may be due to the presence of all these phenolics and others and consequently, it could be applied in different industry sectors, such as food, cosmetic, and pharmaceutical ones.

## 4. Conclusions

BSG contains an important amount of phenolic compounds with antioxidant activity, which can be recovered by solid-to-water extraction. The use of water would allow reducing the cost of extraction in comparison to using organic solvents and would be environmentally friendly. Further, 30 °C, 121.9 min, and 10 mL/g liquid/solid ratio were the most efficient conditions in order to extract antioxidant phenolic compounds. In optimal extraction conditions, simple phenolic acids, such as gallic acid, were the most present phenols. Further research should be undertaken in order to assay the effectiveness of these extracts when added into food.

## Figures and Tables

**Figure 1 foods-09-01398-f001:**
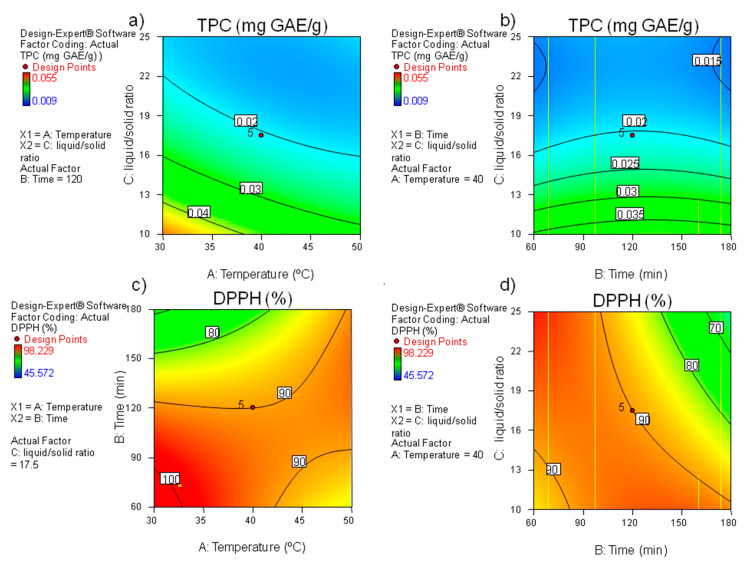
Contour plots for the effect of extraction variables on (**a**,**b**) TPC (mg gallic acid/g BSG) and (**c,d**) DPPH values (% inhibition).

**Figure 2 foods-09-01398-f002:**
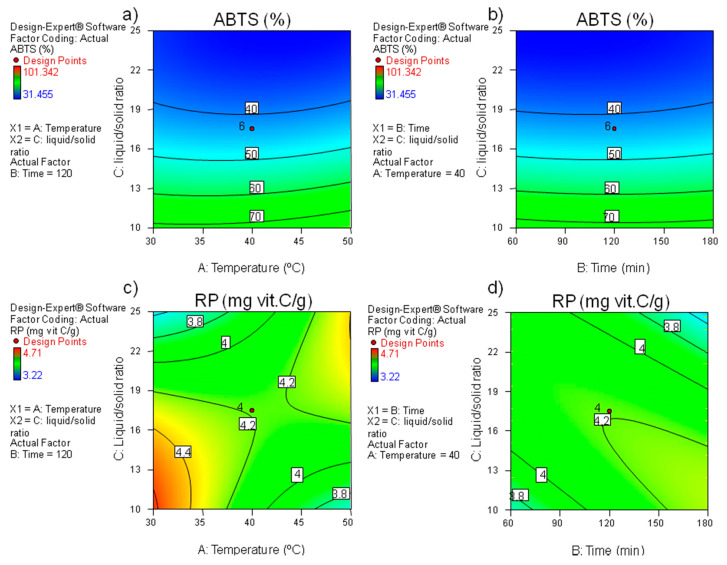
Contour plots for the effect of extraction variables on (**a**,**b**) ABTS values (% inhibition) and (**c**,**d**), RP values (mg ascorbic acid/g BSG).

**Table 1 foods-09-01398-t001:** Experimental factors and measured values of responses. The extraction factors were temperature in °C (*X*_1_), time in minutes (*X*_2_), and ratio *v*/*w* (mL/g) (*X*_3_). The responses were measured in triplicate as total phenol content (TPC) expressed as mg gallic acid/g BSG and antioxidant activity, by DPPH, RP, and ABTS methods expressed as % inhibition, mg ascorbic acid/g BSG, and %, respectively. Values are reported as mean ± SD (*n* = 3).

	Independent Variables			Responses			
Run	Temperature (*X*_1_)	Time (*X*_2_)	Ratio *v*/*w* (*X*_3_)	TPC	DPPH	RP	ABTS
1	40	120	17.5	2.20 ± 0.08	86.38 ± 1.41	12.50 ± 0.51	46.65 ± 1.12
2	50	60	10	3.01 ± 0.06	77.56 ± 2.35	12.09 ± 0.48	78.58 ± 2.43
3	30	60	25	1.51 ± 0.02	39.83 ± 1.21	13.85 ± 0.55	34.25 ± 1.43
4	40	220.91	17.5	2.01 ± 0.05	62.10 ± 1.88	13.85 ± 0.55	46.75 ± 2.13
5	30	180	25	0.91 ± 0.02	45.57 ± 1.38	13.22 ± 0.53	35.96 ± 1.43
6	50	180	10	3.61 ± 0.06	98.23 ± 2.98	13.96 ± 0.56	81.43 ± 1.90
7	23.18	120	17.5	3.81 ± 0.07	96.51 ± 2.92	14.50 ± 0.58	50.56 ± 1.55
8	40	19.09	17.5	1.11 ± 0.03	92.49 ± 2.80	13.90 ± 0.56	44.55 ± 2.23
9	40	120	17.5	1.21 ± 0.04	93.61 ± 2.84	14.46 ± 0.58	48.42 ± 2.27
10	50	180	25	1.51 ± 0.04	84.49 ± 2.56	14.17 ± 0.57	37.04 ± 1.65
11	40	120	30.11	2.91 ± 0.06	84.70 ± 2.57	13.26 ± 0.53	34.30 ± 1.56
12	40	120	17.5	2.51 ± 0.05	96.67 ± 2.93	14.31 ± 0.57	47.72 ± 1.88
13	56.82	120	17.5	1.71 ± 0.03	95.49 ± 2.89	14.24 ± 0.57	56.15 ± 2.53
14	50	60	25	1.71 ± 0.03	85.78 ± 2.60	14.71 ± 0.59	33.76 ± 1.03
15	30	60	10	5.11 ± 0.011	95.87 ± 2.91	14.29 ± 0.57	73.97 ± 2.36
16	40	120	17.5	2.41 ± 0.05	87.60 ± 2.65	13.92 ± 0.56	39.72 ± 1.81
17	40	120	4.89	5.51 ± 0.01	93.77 ± 2.84	13.80 ± 0.55	101.34 ± 5.511
18	30	180	10	4.21 ± 0.08	90.50 ± 2.74	14.71 ± 0.59	69.89 ± 3.18
19	40	120	17.5	2.51 ± 0.05	79.92 ± 2.42	14.11 ± 0.56	31.45 ± 1.23
20	40	120	17.5	1.61 ± 0.02	91.79 ± 2.78	13.12 ± 0.52	43.37 ± 2.97

**Table 2 foods-09-01398-t002:** Regression coefficient (β), coefficient of determination (*R*^2^), and F-test values of the predicted second order polynomial models for TPC, DPPH, RP, and ABTS.

Regression Coefficients (β)				
	TPC	DPPH	RP	ABTS
Intercept				
*X* _0_	2.057	89.999	14.194	45.130
Linear				
*X* _1_	−0.397 *	0.588	−0.029	0.682
*X* _2_	0.030	−7.021 **	−0.010	0.546
*X* _3_	−1.074 **	−4.189	−0.087	−20.181 ***
Cross product				
*X* _12_	0.237	10.443 **	0.046	1.060
*X* _13_	0.437	3.862	0.448 **	−1.946
*X* _23_	−0.062	−9.423 **	−0.286 **	0.778
Quadratic				
*X* _1_ ^2^	0.209	1.709	0.084	1.366
*X* _2_ ^2^	−0.214	−4.905 *	−0.090	0.410
*X* _3_ ^2^	0.722 **	−0.682	−0.212 **	8.248 **
*R* ^2^	0.89	0.88	0.90	0.99
*F* value (model)	7.91 **	6.47 **	7.13 **	66.19 ***
*F* value (lack of fit)	1.21	1.22	0.54	0.72

Level of significance * *p* < 0.05; ** *p* < 0.01; *** *p* < 0.001. *X*_1_ = Extraction temperature (°C), *X*_2_ = Extraction time (min), *X*_3_ = Liquid/solid ratio (*v*/*w*), TPC = Total phenolic content (mg gallic acid/g BSG), DPPH = 2,2-diphenyl-1-picrylhydrazyl radical scavenging ability expressed as inhibition percentage (%), RP = Ferric reducing antioxidant power (mg Vitamin C/g BSG), ABTS = 2,2′-azino-bis(3-ethylbenothiazoline-6-sulphonic acid) radical cation percentage inhibition (%), *R*^2^ = Coefficient of determination.

**Table 3 foods-09-01398-t003:** Pearson’s correlation coefficient calculated for the response variables.

	TPC	DPPH	RP	ABTS
TPC	1	0.434	0.179	0.788 **
DPPH		1	0.513 *	0.402
RP			1	0.155
ABTS				1

Level of significance * *p* < 0.05; ** *p* < 0.01; TPC = Total phenolic content (mg gallic acid/g BSG), DPPH = 2,2-diphenyl-1-picrylhydrazyl radical scavenging ability expressed as inhibition percentage (%), RP = Ferric reducing antioxidant power (mg Vitamin C/g BSG), ABTS = 2,2′-azino-bis(3-ethylbenothiazoline-6-sulphonic acid) radical cation percentage inhibition (%).

**Table 4 foods-09-01398-t004:** Experimental data of the validation of predicted values at optimal extraction conditions.

Dependent Variables	Predicted Value	Experimental Value	% CV
TPC	4.89	5.42	8.3
DPPH	98.23	86.26	9.2
RP	14.64	13.93	11.6
ABTS	72.26	71.58	0.6

TPC = Total phenolic content (mg gallic acid/g BSG), DPPH = 2,2-diphenyl-1-picrylhydrazyl radical scavenging ability expressed as inhibition percentage (%), RP = Ferric reducing antioxidant power (mg Vitamin C/g BSG), ABTS = 2,2′-azino-bis(3-ethylbenothiazoline-6-sulphonic acid) radical cation percentage inhibition (%), % CV = Coefficient of variation.

**Table 5 foods-09-01398-t005:** Phenolic profile of BSG extract obtained in optimal conditions (results were expressed on a dry matter basis).

	Phenolic Compounds (µg/g)
Epicatechin	28.91 ± 2.90
Gallic acid	83.30 ± 9.58
Protocatechuic acid	15.72 ± 5.79
4-Hydroxy benzoic acid	4.04 ± 2.44
4-Hydroxyphenylacetic acid	32.40 ± 5.13
Vanillic acid	1.07 ± 0.73
Caffeic acid	1.69 ± 1.65
Syringic acid	2.51 ± 1.61
Coumaric acid	0.27 ± 0.06
Ferulic acid	0.88 ± 0.11
Sinapic acid	7.16 ± 2.29

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
