# Peer review of "Optimization of Extraction Conditions to Improve Phenolic Content and In Vitro Antioxidant Activity in Craft Brewers’ Spent Grain Using Response Surface Methodology (RSM)"

_foods, 2020, doi:10.3390/foods9101398_

Round 1
Reviewer 1 Report
All my comments and suggestions have been incorporated in the revised manuscript.
Author Response
Authors are grateful for reviewer 1´s comments .
There are no author´s notes to reviewer 1.

Reviewer 2 Report
The work is surely improved after the first reviewing cycle. I have seen that the Authors fully discussed and replied to the issues raised by the reviewers.
Unfortunately, some minor revisions are required.
- line 24. please correct “protocatechuic” (see also Table 5 and line 466).
- line 25. please correct “4-hydroxy benzoic” (see also line 466).
- line 30. please correct to “efficient and simple”.
- line 87. vegetable are intended as products from horticulture. Could it be changed to the more specific “crop by-products substrates”?
- lines 88-94. As regards my comment regarding the sentence in line 78, I asked the Authors, for readers not exactly experienced with advanced statistic methods (like myself), to add some information, and possibly, a literature reference on the used Box-Behnken (B-B) design, a type of “Response Surface Methodology, (RSM)”, and briefly explain the connection between the two term, since the latter has been used in the title. If I well understood, the B-B design is a procedure to perform the RSM, but, in this case the CCD approach was used. Now, very clear this paragraph to explain the concept.
- line 121. correct “immediately”.
- line 314. correct “dissolved”.
- line 369. correct “established”.
- line 500. In this point, I should add a brief sentence on the phenolic composition of these antioxidant extracts, such as “In optimal extraction conditions, simple phenolic acids, such as gallic acid, were the most present phenols”.
- line 501. correct “effectiveness”.
Author Response
Authors are grateful for reviewer 2´s comments and are willing to carry out the minor revisions indicated:
- line 24. “protocatechuic” has been corrected in line 24, table5 and line 466(see also Table 5 and line 466).
“protocatechuic” has been corrected in line 24, table 5 and line 466.
- line 25. please correct “4-hydroxy benzoic” (see also line 466).
“4-hydroxy benzoic” has been corrected in line 5,Table 5, line 364, and line 369
- line 30. please correct to “efficient and simple”.
It has been corrected as indicated in line 30
- line 87. vegetable are intended as products from horticulture. Could it be changed to the more specific “crop by-products substrates”?
¨vegetable¨ has been substituted by ¨crop by-product substrates¨ in line 81 since the reference allows it.
- lines 88-94. As regards my comment regarding the sentence in line 78, I asked the Authors, for readers not exactly experienced with advanced statistic methods (like myself), to add some information, and possibly, a literature reference on the used Box-Behnken (B-B) design, a type of “Response Surface Methodology, (RSM)”, and briefly explain the connection between the two term, since the latter has been used in the title. If I well understood, the B-B design is a procedure to perform the RSM, but, in this case the CCD approach was used. Now, very clear this paragraph to explain the concept.
Box-Behnken design was not used in the current experiment. Central Composite Design was. This is explained in line 82-88.
A response surface design is a set of planned experiments that are useful to understand and optimize the response (dependent variables, in this case, TPC and antioxidant assays values). A great number of designs can be used depending on the type of experiment, variables, aim, etc….such as Box-Behnken or Central Composite Design.
CCD has been selected since it shows a higher number of design points (runs), each factor can show up to 5 levels and runs where all factors are at their extreme settings are can be also included. In summary, CCD best fitted our experiment.
- line 121. correct “immediately”.
Iine 101, “immediately¨ has been corrected
- line 314. correct “dissolved”.
Line 262, “dissolved” has been corrected
- line 369. correct “established”.
Line 306, “established” has been corrected
- line 500. In this point, I should add a brief sentence on the phenolic composition of these antioxidant extracts, such as “In optimal extraction conditions, simple phenolic acids, such as gallic acid, were the most present phenols”.
The indicated sentence has been included in line 393
- line 501. correct “effectiveness”.
Line 395, “effectiveness” has been corrected

Reviewer 3 Report
Dear authors,
Thank you for sending us the revised version of your manuscript. I am still not fully understand why vitamin C was chosen for the calibration curve, i.e for DPPH, reducing power and ABTS. Simply citing other publications for using vitamin C as standard is not right. The reason for using gallic acid is because grain in general is rich in phenolic compounds and not vitamin C. Please find examples of the relevant info here: https://pubmed.ncbi.nlm.nih.gov/?term=spent+grain+phenolic+
Vitamin C is more suitable for fruits and vegetables.
I would be grateful if you can improve this section.
All the best and thank you.
Author Response
English changes required have been carried out all throughout the text.
We understand reviewer perspective. However, ascorbic acid as standard can be also considered as correct from our experience. Ascorbic acid together with other powerful antioxidants such as trolox or EDTA, have been widely used as standards to determine antioxidant activity in food regardless composition of matrix food. For example, olive oil is very rich in phenolic compounds but trolox and EDTA have been used for the calibration curves (Peršurića et al., 2020). Antioxidant assay results express the antioxidant activity of samples in a way that this activity is equivalent or similar to the activity of a certain amount of acknowledged antioxidant compounds such as ascorbic acid, trolox or EDTA.
In the study from Yu et al. (2002), the radical DPPH scavenging capacities of the 4 cereal products were measured and compared among each other as well as they were also compared to α-tocopherol, ascorbic acid and BHT. Ferri et al. (2013) optimised different methodologies in order to measure free and bound components of the antioxidant capacity of wheat flour using both methods, ABTS and DPPH, the ascorbic acid being used as standard.
However, considering reviewer 3´s comments on this issue, we will consider using Gallic acid in the forthcoming experiments.
Peršurića, Z., Martinovića, L.S., Zengin, G., Šarolić, M. Pavelića, S.K. (2020) Characterization of phenolic and triacylglycerol compounds in the olive oil by-product pâté and assay of its antioxidant and enzyme inhibition activity. LWT - Food Science and Technology,125, 109:225. https://doi.org/10.1016/j.lwt.2020.109225.
YU, J. PERRET, B. DAVY, J. WILSON, AND C.L. MELBY (2002). Antioxidant Properties of Cereal Products. Journal of Food Science 67, 2600-2603. https://doi.org/10.1111/j.1365-2621.2002.tb08784.x
Ferri, M., Gianotti, A., Tassonia, A. (2013). Optimisation of assay conditions for the determination of antioxidant capacity and polyphenols in cereal food components. Journal of Food Composition and Analysis, 30, 94-102. https://doi.org/10.1111/j.1365-2621.2002.tb08784.x

Reviewer 4 Report
Table 2; in Cross Product; Please change
X12 with X12
X13 with X13
X23 with X23
Author Response
Table 2; in Cross Product; Please change
X12 with X12
X13 with X13
X23 with X23
The indicated changes have been written in table 2

This manuscript is a resubmission of an earlier submission. The following is a list of the peer review reports and author responses from that submission.
Round 1
Reviewer 1 Report
The manuscript is well written and it highlights important scientific point. However, there are issues in the manuscript that should be addressed.
- The optimized time for the extraction is confounding. In table 1, it is clear to see TPC is highest for run 15, moreover other observed responses also have higher value.
In line 222 author wrote extraction time did not play any significant contribution towards TPC. While in conclusion line 343 authors selected parameter with higher run time (121,9 min)? Please clarify the selection of these parameter?
- Line 78: Box-Behnken design is mentioned in the line 78 to be used for the experiment design. However, Central Composite Design was actually used for the present study. Correct the sentence accordingly.
(Actually if BBD was used then that would have 12 run + central points with no axial point for 3 factors.)
- Table 2: Please replace Quadratic X12, X22 and X32 with X12 X22 and X32.
- Table 2 Please describe the abbreviation T, D R and ABT?
- Figure 1: The contour plot is unclear. Hard to interpret the result. Please increase the font size in a readable manner.
- Please provide the HPLC chromatogram of BSR extract and Standards calibration curve as a supplementary file for the reader.
Reviewer 2 Report
General Comments:
The use of vitamin C for antioxidant activity was not appropriate. Should have used gallic acid for ‘general’ TPC determination.
Title:
Change the word ‘for’ to ‘to’…improve…
Abstract:
Line 17: Please add some detail on the results.
Line 18: Sentence….’30ºC, 121,9 min and 10 mL /g liquid/solid’ What do you mean by this sentence? 121.9 min? Also the symbol for degree Celsius should not be underlined.
Introduction:
Line 27: Write in full hL. Ref as super script 1? Please check this.
Methodology:
Line 83: Please use ‘dot’ instead of comma
Line 140: Change ‘were’ to ‘are’
Line 149: It was not appropriate to use vitamin C to use vitamin c as positive control. Should use gallic acid at least. Also include the range of standards used to prepare the calibration curve. Please add new data for this section.
Line 153: As per comment on Line 149. Should have used gallic acid.
Line 154: Check spelling
Line 166: Ae per comment on Line 149. Should have used gallic acid.
Line 184: Check grammar
Results and Discussion:
General comment: Please use ‘dot’ instead of comma.
Line 226: The figure legends are blurred. Please change this.
Line 286: Poor correlation could be due to the wrong ‘standard’ used for TPC analysis.
Line 318: There were huge differences between TPC (Table 4, mg per g) and individual phenolics (Table 5, μg per g). Again, this could be due to the wrong ‘standard’ used to prepare the calibration curve. Please add new data for this section.
Line 321: Vitamin C not vitamina C.
Line 335: Table 5 - μg/g per g of dry or wet basis. Please specify.
Conclusions:
Need to be careful with the conclusion since the ‘standard’ used was vitamin C, which is not phenolic. Also there were poor correlation between TPC and DPPH and RP.
Reviewer 3 Report
The manuscript is on the extraction optimization from BSG using RSM.
As authors mentioned, the extraction procedure is too simple. There is not sufficient reason for optimization using RSM.
Also, there was already known that the relationship between total phenolic compounds and antioxidative acitivities. Optimization of total phenolic compounds will be enough.
Reviewer 4 Report
The work deals with the optimization of a method to extract the antioxidants from spent grain to make beer. The main comment I see in the present work is that no report has been made about the type of solvent (is it water, from Line 20?) to extract phenolics from the matrix. This should be cited in the Abstract, in order to improve the applicability of the findings.
The second main problem is that in the present work the treatment regards the phenolics in their free form. It is well noted that many phenols, especially in grains, are under bound form. This issue should be stressed over the manuscript, for example in Line 42, enforced by literature.
Finally, a deep discussion regarding the single phenol composition of the optimized extract is needed, possibly with the better comparison with recent literature data.
The conclusion is that the work, to be considered for publication, needs some major amendments. Moreover, a careful check overall the entire manuscript has to be done, since several typing errors are present. Here below are listed the specific comments to the manuscript.
Lines 42-47: in this paragraph, some sentence related to the type of phenols interesting for this recovery should be added. To my experience, these are mainly belonging to the class of acidic phenols (vanillic, ferulic acid, and so on).
Line 61: not at all. Often, the best solvent for phenols extraction is a binary mixture of water and EtOH and/or MeOH. Please, clarify, also with literature.
Line 78: please, explain with a literature reference.
Lines 81-87: what type of drier was used? Normal oven? Convective oven? Else? Please, specify. Was the dried material used as is, or it was reduced in powder? In each case, another quality index of the used spent grain to be possibly added should be the particle size.
Lines 95-100: please, give a value regarding the used amounts of dried matter in the experiments.
Line 101, Table 1: I see some strange values in the T and Time columns (23.18, 56.82, and so on). Are they typing errors? Or are values given from the algorithm? Please, clarify.
Line 108: how was the pH stabilized at the given value? Was it buffered? Please, explain.
Lines 139-141, and Line 295: in this case, it should be useful to remember that the Index of Folin-Ciocalteu could be not only related to phenols, but also be the result of other reduction reactions, such as by ascorbates and thiols. This can explain also the lack of correlation of phenols content with antioxidant assays of the present experiment. Literature enforces the present comments: for example see Everette et al, J. Agric. Food Chem. 2010, vol 58(14): 8139–8144, doi, https://doi.org/10.1021/jf1005935.
Lines 144-149: to better explain the antioxidant methods in which the results are given in percent, the final concentration of the radical should be given, as well as the used amount of the extract in the assay.
Lines 174-184: I see that, interestingly, the matrix characterization at optimum conditions was made by HPLC. Why this important issue was not reported in the Abstract and in the Introduction, reporting and enforcing by literature the phenolic composition of spent grain beer?
Line 332: infact, the phenols present in minor amount in the extract of the present experiment (p-coumaric, caffeic and ferulic acids) have a low solubility in water. Their extraction is enhanced by the use of organic solvents, so confirming the relevant difference with other reports.
Line 334: the present material to be extracted is not only from wheat, but also from barley (see line 82). This increases the difference with previous works.
Line 335: from the Table 5, I see that a relevant amount is related to "hydroxy phenolic acid" . The Authors should better explain this type of compound, since it is not clear. Just as an example, syringic acid is an hydroxy phenolic acid, specifically, the 4-hydroxy-3,5-dimethoxybenzoic acid. Please, clarify.
Reviewer 5 Report
The aim of the article was optimize the factors such as temperature, time and liquid/solid ratio on the phenolics content in extracts. The topic is not innovative because there are many publications on this subject. However, the presented experiment is multifactorial and the application of the Box-Behnken method allows you to find out what extraction conditions should be used to obtain an aqueous extract with a high content of polyphenols. This has practical significance that can be used in the food or pharmaceutical industry. Analytical methods are correct, generally used, the references are adequate.
Detailed comments:
What polyphenolic compounds are determined in the method with the Folin reagent? The article suggests that compounds like flavonoids, tannins, proanthocyanidins and amino phenolic are not determined with this spectrophotometric method (L297). In that case, reference should be made to the literature showing that the Folin method does not determine for example: flavonoids.
Table 5 shows that the total content of polyphenolic compounds determined in the extract is over 0.177 mg per g. However, the total content of polyphenols for an extract obtained under similar conditions is approximately 5 mg per g according to Table 1. Why were such differences observed? What substances may the tested material contain, that the content of polyphenols by spectrophotomeric method and HPLC is so different. Could protein substances have an influence on the determination of total polyphenols? I recommend that these aspects be discussed.
In the article, the spelling of some words should be corrected: catión, acdi (Tabele 5), tensión, máximum
Figure 1 is illegible. The Y axis in the charts is not readable. The values in the squares marked in the graphs are not visible. There is also no figure legend. What do the colors in each figure mean?
The article does not explain why the antioxidant capacity was determined by three different methods. Why were these three methods used? Is it just because they are the most popular methods of determining the antioxidant capacity? It has been suggested that other compounds have little effect on the measured capacitance. What are these relationships? This aspect should be discussed in more detail.